# Moderate Caloric Restriction Partially Improved Oxidative Stress Markers in Obese Humans

**DOI:** 10.3390/antiox10071018

**Published:** 2021-06-24

**Authors:** Dominika Kanikowska, Alina Kanikowska, Ewelina Swora-Cwynar, Marian Grzymisławski, Maki Sato, Andrzej Bręborowicz, Janusz Witowski, Katarzyna Korybalska

**Affiliations:** 1Department of Pathophysiology, Poznan University of Medical Sciences, 60-806 Poznań, Poland; abreb@ump.edu.pl (A.B.); jwitow@ump.edu.pl (J.W.); koryb@ump.edu.pl (K.K.); 2Department of Internal Diseases, Metabolism and Nutrition, Poznań University of Medical Science, 60-355 Poznań, Poland; akanikowska@ump.edu.pl (A.K.); eswora@ump.edu.pl (E.S.-C.); mariangrzym@ump.edu.pl (M.G.); 3Department of Physiology, Institutional Research, Aichi Medical University School of Medicine, Aichi Medical University, Aichi 480-1195, Japan; msato@aichi-med-u.ac.jp

**Keywords:** oxidative stress, obesity, moderate calorie restriction, MPO

## Abstract

Oxidative stress and inflammation are implicated in obesity. Therefore, we investigated whether moderate and short-term calorie restriction (CR) reflects a real-life situation, mediates weight loss, and improves oxidative stress markers. We analyzed oxidative stress markers in patients with obesity undergoing moderate CR. Serum oxidative stress markers (myeloperoxidase (MPO), superoxide dismutase (SOD), catalase, total antioxidant status (TAS), and reactive oxygen species (ROS) (generation by endothelial cells in vitro)) were measured in 53 subjects (mean BMI 37.8 ± 5.9 kg/m^2^) who underwent 8 weeks of CR, which included a reduction of 300–500 kcal/day. MPO was the most CR-sensitive parameter. The mean level of serum MPO in patients with obesity was 20% higher than that in post CR intervention (*p* < 0.001). SOD increased by 12% after CR (*p* < 0.05), which was largely due to the improvement in glucose tolerance and the reduction in insulin resistance after CR. Other tested parameters were not modified during the treatment. CR resulted in an expected decrease in body weight (by 5.9 ± 4.6 kg, *p* < 0.0001) and other anthropometric parameters. Additionally, it was accompanied by a significant change in hsCRP, hsTNF alpha, hsIL-6, leptin (all *p* < 0.0001), and HOMA-IR (*p* < 0.05). Cardiovascular and metabolic parameters were also partially improved. Short-term, moderate CR partially improves antioxidant capacity but is enough to substantially change anthropometric parameters in obese patients. Our observations indicate that mimicking real-life situations and low-cost dietary intervention can be successfully implemented in obesity treatment with a simultaneous moderate effect on antioxidant status.

## 1. Introduction

Being obese is increasing in frequency in many countries and is associated with several problems, including metabolic disorders or syndrome [1]. Obesity is associated with a persistently increased inflammatory state [2]. The expansion and dysfunction of adipose tissue produces several bioactive substances, which generate chronic low-grade inflammation, contributing to obesity-related metabolic diseases [3]. Chronic inflammation intensifies oxidative stress (OS) by accumulating reactive oxygen species (ROS) and reducing antioxidant defence [4]. Inflammation and oxidative stress are complications associated with obesity and are related to the genesis of chronic metabolic diseases [5,6].

Several mechanisms are involved in generating OS in obesity, including adipokine initiation to produce ROS, and the mitochondrial and peroxisomal oxidation of fatty acids, which can produce ROS. In contrast, another mechanism is the over-consumption of oxygen, which generates free radicals in the mitochondrial respiratory chain [7].

Of note, obesity could, at least partly, be reversed by targeted lifestyle improvements, including physical activity and nutrition [8]. Lifestyle modifications, a healthy diet, and an increase in exercise are beneficial for preventing and treating obesity [9].

Although the association of obesity with increased inflammatory and decreased antioxidant capacity has been well documented [10,11], only a few studies examined the effects of weight-loss regimens on oxidative stress [12,13].

Lopez-Domenech et al. (2018) found that a very low-calorie diet for six months in morbidly obese patients improved anthropometric and biochemical parameters, ameliorated the inflammatory response, decreased pro-oxidant agents such as myeloperoxidase (MPO), and increased antioxidant capacity such as catalase activity [12].

However, most obese subjects may have difficulties in adhering to prolonged and substantial caloric restriction, resulting in insignificant weight loss, failure to maintain any long-term diet regimen, and eventual regain of body weight [14]. In this study, we addressed this issue in a setting more likely to reflect a real-life situation and assessed inflammatory and ROS parameters in obese volunteers who were undergoing very short-term (8 weeks) moderate calorie restriction (15–30% energy deficit).

We investigated possible changes in serum antioxidant capacity defending against ROS in patients with simple obesity undergoing moderate calorie restriction. Our previous study [15] showed that patients with obesity comply well with such a treatment regimen. Furthermore, we observed that even short-term dietary intervention of this kind can produce biochemical changes that are supposed to be beneficial for metabolic health. This lead to the question of whether these included changes in serum antioxidant properties and whether they correspond to the decreased body weight.

## 2. Methods

### 2.1. Participants and Places

The study protocol was approved by the Ethics Committee of Poznan University of Medical Sciences (No. 217/11), and all participants gave their informed consent. The study adhered to the basic principles of the Declaration of Helsinki.

The subjects were individuals with obesity with BMI > 30 kg/m^2^ (*n* = 53). Patients with obesity were recruited from the University Department for Internal Diseases, Metabolism and Nutrition (Poznań University of Medical Sciences, Poznań). The inclusion/exclusion criteria are listed in Table 1.

Initially, we invited eighty-four patients with BMI > 30 kg/m^2^ from a group of patients willing to undergo dietary intervention as a means of losing weight. However, the whole dietetic restriction of the study was completed by only fifty-eight participants (drop-out rate: 31%). Five volunteers were excluded from the initial group because of gaining weight (min: +3.5 kg, max: +13.5 kg), and not following the dietary regimen (drop-out rate 0.08%). Fifty-three patients who completed the eight week calorie restriction (CR) diet, the great majority, lost weight (*n* = 50; weight loss: min: (−) 1.5 kg, max: (−) 13.5 kg, average: (−) 5.9 kg) and during this time had stable body mass or gained no more than 1 kg (*n* = 3 min: (+) 0.1 kg, max: (+) 0.8 kg). The eight week dietary intervention was performed at home. Patient characteristics are given in Table 2. The intervention consisted of a moderately low-calorie diet. A reduction in the daily calorie intake was 15–30% (a reduction by 300–500 kcal/day), compared to the patients’ calculated basal metabolic rate (BMR) by the Harris and Benedict formula [16] and the physical activity rate corrected according to WHO criteria. The estimated BMR ranged between 1616 and 2954 kcal/d (average: 1965 kcal/day) [16], and all patients displayed low physical activity (physical activity factor: 1.4) [17]. To accurately determine total energy expenditure and prepare an appropriate diet, the daily dietary intake was assessed in all subjects over 3 days prior to entering the program. Subjects were advised to take a similar diet under the guidance of the supervising dietician, who designed individualized dietary plans. That supplied energy from similar sources (25% fat, 20–25% protein, 50–55% carbohydrates) but took into account patients’ food preferences (detailed diet content is listed in Table 3). Physical activity did not accompany the diet and was not controlled during the treatment. If required, an oral hypoglycaemic agent (metformin) was introduced at the discretion of an attending physician. Patients with glucose intolerance (and hyperinsulinemia) were treated with metformin (*n* = 15). Additionally, metformin was also introduced at the physician’s discretion to other obese patients (*n* = 9). Finally, 24 participants were treated with metformin (Table 2). Other medications for treating chronic diseases were allowed as long as they did not change throughout the study. Glucose intolerance and hypertension are very common abnormalities seen in obese patients; therefore, we decided to include them in our research. 

Before and after the intervention, the patient’s weight and body composition was measured by bioimpedance spectroscopy (Tanita MC 980 MA, Tokyo, Japan), and blood samples were collected. 

### 2.2. Biochemical Measurements

Samples of serum were collected by routine methods at the same time of day (07:30–09:00) to avoid circadian variations and stored at −80 °C until assayed in batch.

Adipokines were measured using R&D Systems (Minneapolis, MN, USA), DuoSets (leptin, adiponectin, vaspin), and quantikine high sensitive assays were used for interleukin 6 (Il-6) and tumor necrosis alpha (TNF-alpha). The sensitivity of analyzed adipokines were as follow: 24.8 pg/mL for leptin, 24.4 pg/mL for adiponectin, 13.9 pg/mL for resistin, 10.4 pg/mL for vaspin, 0.11 pg/mL for hs TNF- alpha, 0.04 pg/mL for hs Il-6. 

The C-reactive protein (CRP) was measured using a high sensitivity assay from BioVendor (Brno, Czech Republik). The sensitivity of the assay was 0.02 µg/mL.

The myeloperoxidase activity (MPO) was measured using R&D Systems (Minneapolis, MN, USA) DuoSet with a sensitivity of 36.9 pg/mL.

The activities of superoxide dismutase (SOD), catalase, and total antioxidant status (TAS) were measured using Cayman tests (Cayman, MI, USA). The TAS test measures the total antioxidant capacity of serum. Aqueous- and lipid-soluble antioxidants were measured, including vitamins, proteins, lipids, glutathione, uric acid, etc. 

The assays’ sensitivity is as follows: below 0.025 U/mL for SOD, below 2 nM/min/mL for catalase, and below 0.045 mM for TAS.

All other measurements were performed by the central laboratory of the university hospital.

The oral glucose tolerance test (OGTT) was performed in all obese patients before and after CR. Glucose intolerance was diagnosed when glucose concentration was ≥140 mg/dL in the OGTT test at baseline (after 120 min). The achieved results of glucose and insulin allowed us to calculate the homeostasis model assessment (HOMA)—an index of insulin resistance (fasting insulinemia (mU/mL) × fasting glycemia mg/dL)/405) [18]. HOMA-IR was calculated before and after CR based on data available from OGTT.

### 2.3. In Vitro Study

#### 2.3.1. Cell Culture

The tests were performed using human umbilical vein endothelial cells (HUVECs) line EA.hy926 (kindly provided by Dr CJ Edgell, University of North Carolina, Chapel Hill, CA, USA) [19]. The HUVECs were cultured in Earl’s-buffered M199 culture medium, supplemented with L-glutamine (2 mmol/L), antibiotics (amphotericin (2.5 μg/mL) and gentamycin (50 μg/mL)), hydrocortisone (0.4 μg/mL), and 10% *v/v* fetal calf serum Invitrogen (Waltham, MA, USA). The cultures were maintained at 37 °C in a humidified atmosphere of 95% air and 5% CO_2_. All the reagents were purchased from Sigma-Aldrich (Burlington, MA, USA). Cell culture plastics were bought from Nunc (Roskilde, Sjælland, Denmark) and Corning-Costar (Glendale, AZ, USA). 

#### 2.3.2. Detection of Reactive Oxygen Species (ROS)

Endothelial cells (2 × 10^4^) were exposed for 24 h on a standard medium supplemented with 10% *v/v* of serum taken from obese patients before and after dietary regimen. Afterward, the medium was discarded, and ROS generation was detected by 2′,7′-dichlorodihydrofluorescein diacetate (H2DCFDA); (Molecular Probes, Eugene, Oregon, USA) labeling. Briefly, cells were incubated with 10 μM H2DCFDA for 45 min at 37 °C and then solubilized with the lysis buffer (Promega, Medison, WI, USA). ROS generation in cells is the degree of fluorescence of the H2DCFDA dye, oxidized by the intracellular ROS to the fluorescent product. Fluorescence emitted by cell lysate was measured using two wavelengths; 485 and 535 nm for excitation and emission, respectively, in a spectrofluorometer (Thermo Fisher Scientific, Waltham, MA, USA). Hydrogen peroxide (100 µM) was run during the assay as the positive control. The data were expressed as a percentage of control. The control consisted of cells cultured in a standard medium supplemented with fetal calf serum (=100%). The experiment was repeated twice. Each patent’s serum before and after CR was tested in duplicate. 

### 2.4. Statistics

Statistical analysis was performed using GraphPad Prism TM 8 software (Software Inc., San Diego, CA, USA). The results are presented as means (±SD). The normality of the distribution was tested using Shapiro–Wilk’s test. The data were analyzed using the paired/nonpaired *t*-tests depending on Gaussian/non-Gaussian distribution (Gaussian distribution: the paired or unpaired test; non-Gaussian distribution: Wilcoxon test or Mann–Whitney test and categorical data were analyzed with the χ^2^ test. All results were considered significant at *p* < 0.05. 

## 3. Results

### 3.1. Description and Distribution of the Obese Subjects

Among fifty-three obese patients, females were predominant (72%). One-third of all participants had morbid obesity with a BMI above 40 kg/m^2^. Our study recruited obese patients with a very high BMI value (median 36.8 kg/m^2^). Fifteen patients (28%) had glucose intolerance, usually associated with higher BMI values (Table 2). Twenty-four of the patients (45%) were treated with metformin for eight weeks of follow-up (Table 2). Eight weeks of moderate CR reduced the percentage of morbid obesity among patients (Table 4). Treated hypertension was reported in 28% of patients, and thyroid disease mainly concerned females (Table 2).

### 3.2. Effect of Modest Dietary Interventions on Serum OS Markers and Endothelial Cell ROS Production In Vitro

Eight weeks CR only partially improved serum antioxidant properties (Figure 1 and Figure 2). MPO proved to be the most sensitive to moderate CR, and its reduction (−20%) was paralleled by a diminishing neutrophil count (−7.3%) (Figure 1A,C). Another enzyme modified by CR was superoxide dismutase. Its activity increased significantly by 12% during the treatment (Figure 2A). Catalase (Figure 2B), total antioxidant status (Figure 2C), and ROS production by endothelial cells exposed in vitro to medium supplemented with serum taken from the obese patients before and after CR (Figure 2D) were not changed by moderate CR.

We divided obese patients according to three criteria: (i) loss of fat mass (above (*n* = 28)/ below (*n* = 25) median value: 4,3 kg), (ii) loss of body weight (above (*n* = 27)/below (*n* = 26) median value: 5.9 kg), and (iii) occurrence of glucose intolerance (*n* = 15) or normal glycemia (*n* = 38). We aimed to determine whether MPO and SOD modification are proportional to the loss of fat mass or body weight, or are glucose intolerance dependent. The decline in MPO was independent of whether patients lost high or low-fat mass (Figure 3A), high or low body mass (Figure 3B), and whether the patients had or did not have glucose intolerance (Figure 3C). A similar relationship was tested for SOD (Figure 4), but its improvement (+30%) was only detected in patients with glucose intolerance after CR. The increase in SOD level was largely due to the improvement in glucose tolerance and the reduction of insulin resistance after CR. Patients exhibit glucose intolerance before CR (after 2 h: median:15 g/dL, min: 141 mg/dL, max: 220 mg/dL) and dietary treatment ameliorates it (after 2 h: median: 127 mg/dL, min: 87 mg/dL, max: 189 mg/dL; *p* = 0.0005). Their insulin resistance was also statistically reduced after CR (Before HOMA-IR: median: 5.0, min: 0.8, max: 9.5; After HOMA-IR: median: 3.3, min: 0.6, max: 7.2; *p* = 0.0107).

As documented in Figure 1, the decrease in MPO was connected with a decreased neutrophil count (−7.3%; Figure 1C). Neutrophil decline was only reported in patients who lost significant body weight (−11%; Figure 3E) and in people with normal glucose tolerance (−9.3%; Figure 3F). 

### 3.3. Effect of Modest Dietary Interventions on Anthropometric and Biochemical Parameters

The impact of CR on patients’ anthropometric, cardiovascular, metabolic, and inflammatory parameters is given in Table 4.

An intervention protocol led to an expected (albeit modest) loss of weight, BMI, WC and body fat, with SBP reduction. In participants, the dietary intervention resulted in an average weight loss of −5.9 ± −3.4 kg, which corresponded to a relative decrease in weight ranging from 0.6% to 13.2% (average weight loss: 5.3% ± 3.1%) (Table 4). The dietary regimen also resulted in a significant decrease in total cholesterol and triglyceride concentrations. However, the treatment did not cause a significant change in LDL and HDL cholesterol concentration. Of the glucose metabolism parameters, glucose and insulin remained unchanged, whereas HOMA-IR decreased (*p* < 0.05) (Table 4). There were large fluctuations in insulin levels among the patients (before CR min:3.1, max:37.4; after CR min: 2.5, max: 90.3), and that is probably why the insulin values were not statistically significant. Leptin (*p* < 0.0001) and vaspin (*p* < 0.001) concentration were also markedly reduced after weight loss.

Surprisingly, however, the treatment did not result in a significant change in serum adiponectin and resistin concentrations either. Though the applied dietary intervention was followed by weight loss, patients were still formally classified as obese after treatment. We hypothesized that the magnitude of weight reduction was insufficient to produce a change in the adiponectin and resistin levels.

However, systemic inflammatory markers were altered by weight loss. Concentrations of hs CRP, hs IL-6 and hs TNF alpha decreased markedly after dietary therapy (all *p* < 0.0001) (Table 4).

## 4. Discussion

A primary focus of our work was to determine if calorie restriction diets would decrease oxidative stress. The antioxidant effect of the dietary intervention was measured. MPO was discovered to be the most calorie restriction sensitive among five tested parameters (MPO, SOD, catalase, TAS, ROS production by endothelium). The serum concentration of MPO was reduced after 8 weeks of moderate CR (*p* < 0.001) (Figure 1A). MPO was chosen as one marker of pro-oxidative stress for this study since it is a widely used and easily assessed marker of OS [20].

Our results showed that MPO was markedly reduced after weight loss, despite the fact that its reduction was not related to the proportion of fat mass and body weight loss and was not connected to glucose intolerance (Figure 3A–C). Patients who lost a small amount of fat mass, body weight and had glucose intolerance also exhibited MPO reduction, which was not paralleled by a decrease in neutrophil count (Figure 3D–F.). MPO is derived mainly from neutrophils and monocytes and plays a crucial role in leukocyte-mediated endothelial injury response [21].

The number of neutrophils decreased after weight loss, although the total leukocyte count remained unchanged (Figure 1B–D), suggesting a reduced MPO expression by a decreased number of neutrophils. Conversely, MPO reduction was paralleled by diminishing neutrophil counts, which was mainly observed after marked weight loss (Figure 3E) and in patients without glucose intolerance (Figure 3F), also suggesting a reduced MPO expression by the leukocyte defence system [22].

Available published data indicate that calorie restriction could also reduce WBC count (especially lymphocytes), but this typically occurs after prolonged and intense nutrient restriction [23,24]. In this study, we used only a short duration and moderate calorie restriction regimen, so no effect should be measured on the circulating total WBC count; however, plausibly, the neutrophils were more sensitive to caloric restriction (Figure 1C).

Furthermore, a potential modulatory and supporting effect on sustaining CR is exerted by metformin, an oral hypoglycaemic agent, which displays multidirectional activity (for more details, see [25]. In this study, 45% of participants received metformin therapy, with an apparent effect on oxidative stress and possible reduction in ROS formation [26,27]. As reported by Bonnefont-Rousselot et al., metformin in a pharmacologically relevant concentration was able to scavenge hydroxyl but not superoxide anion in vitro. Their results suggest that metformin could directly scavenge ROS or indirectly modulate superoxide anion production by leukocyte NADPH oxidase, the essence of an oxidative burst. [27]. Chakraborty et al. detected ROS production in WBCs after metformin treatment. They discovered reduced ROS production by WBCs in metformin-treated diabetic patients but not in the placebo group [26]. Metformin, used in 45% of our patients (not only with glucose intolerance), likely also contributed to MPO reduction by modulating ROS production by WBCs and its direct modulation effect on hydroxyl radical superoxide anion production.

SOD catalyzes the superoxide anion to hydrogen peroxide, which, with the chloride anion, is a MPO substrate that creates antibacterial hypochlorous acid.

Additionally, we observed that dietary weight loss also resulted in an increase in antioxidant capacity (SOD) (*p* < 0.05) (Figure 2A), but not in TAS and catalase concentration (Figure 2B,C).

This, in line with other studies by [12,13,28], indicates that caloric restriction increases antioxidants. Following a similar study design [29], SOD concentration increased after 12 weeks of moderate CR in a group of patients with diabetes mellitus type 2. Our results showed that SOD level was increased after weight loss (Figure 2), and its increase was mainly related to improving glucose tolerance and insulin sensitivity (Figure 4). Only patients who had glucose intolerance exhibited SOD increase.

A general link between insulin resistance and oxidative stress has been proposed [30]. It has been shown to support a specific role for mitochondrial superoxide anion in insulin resistance in muscle and fat cells [31]. Hoehn et al. (2009) showed that in insulin-resistant cells, overexpression of SOD led to an overshoot in insulin and stimulated GLUT4 translocation [32]. In contradiction, in a survey by Dai C. et al. (2016), the hyperglycemic condition did not change superoxide and antioxidant enzymes levels, including SOD level [33].

The possible mechanism for modulation of the oxidative stress markers induced by weight reduction could be a reverse mechanism by which obesity produces oxidative stress (including adipokine initiation to produce ROS).

Oxidative stress has been proposed to parallel inflammatory processes [34], so the secondary aim of this study was to detect the effect of moderate caloric restriction on the inflammatory response. Systemic inflammatory markers were altered by weight loss. HsCRP, hsTNF alpha and hsIL-6, which are known to be associated with obesity, decreased markedly after dietary intervention (*p* < 0.0001) (Table 4). Furthermore, leptin, which is elevated in obese people and has also been engaged in a pro-inflammatory and pro-oxidative activity [35], was also reduced (*p* < 0.0001) (Table 4). Diminished leptin concentration could also explain the lowered oxidative stress capacity observed in this study.

Moreover, leptin is involved in blood pressure regulation [36], and obese patients usually have hypertension. Decreased leptin concentration could be engaged in the SBP reduction observed in this study (Table 4).

It is well known that both prolonged and short-lasting calorie restriction leads to an improvement in anthropometric parameters [15,37]. Reducing body weight and fat mass essentially decreases pro-inflammatory adipokines and lipid parameters, and increases insulin sensitivity [38,39]. In the present study, we demonstrated that reducing adipose tissue mass and, consequently, decreasing plasma leptin, hsCRP, hsIL-6 and hsTNF alpha, improves lipid parameters and increases insulin sensitivity.

## 5. Conclusions

In conclusion, this study showed that a moderate CR improves antioxidant capacity, decreases body weight, waist circumference, blood pressure, total cholesterol, triglycerides, chronic inflammation, and improves glucose tolerance and insulin sensitivity. These findings are significant for public health and indicate that inexpensive and safe calorie restriction, based on even short and moderate dietary interventions can be effective in optimizing metabolic health and are worth implementing in obesity treatment. We believe that the long-term benefits of short-term, moderate CR include reducing body weight and concern for good eating habits, improving patients’ health conditions.

## Figures and Tables

**Figure 1 antioxidants-10-01018-f001:**
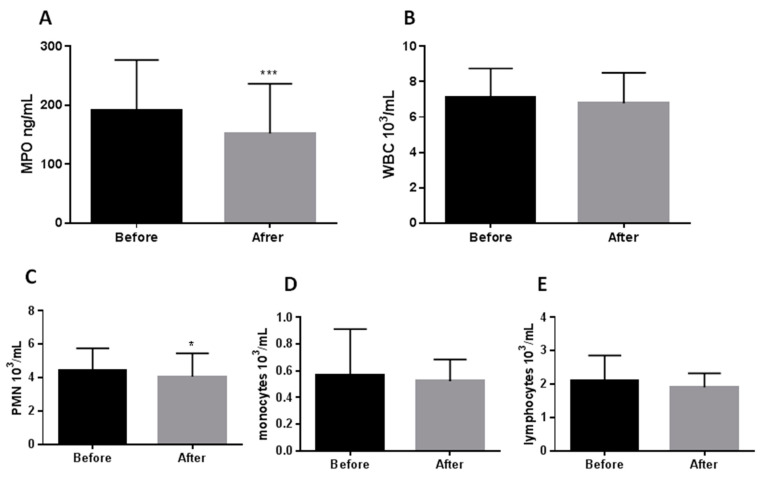
Effect of 8 week caloric restriction on serum myeloperoxidase activity (**A**), total white blood cell count (**B**), neutrophile count (**C**), monocyte count (**D**), and lymphocyte count (**E**). The data were analyzed using the paired *t*-tests (Gaussian distribution: the paired test; non-Gaussian distribution: Wilcoxon test). Significance difference Before vs. After: * *p* < 0.05, *** *p* < 0.001; Abbreviations: WBC—white blood cell count, PMN—polymorphonuclear leukocytes, MPO—myeloperoxidase.

**Figure 2 antioxidants-10-01018-f002:**
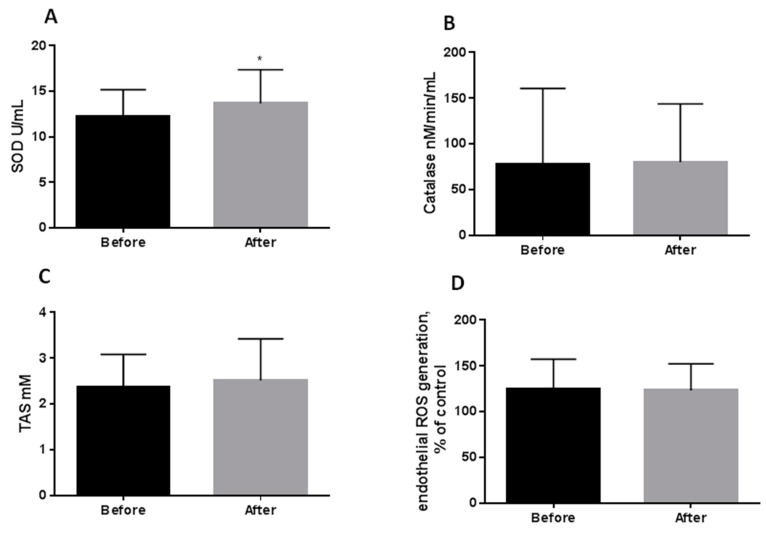
Effect of 8 weeks caloric restriction on serum superoxide dismutase activity (**A**), catalase activity (**B**), total antioxidant status (**C**), and reactive oxygen species generation by endothelial cells cultured in vitro (*n* = 2) in medium supplemented with serum taken from obese patients before and after eight weeks of caloric restriction (**D**). The data were analyzed using the paired *t*-tests (Gaussian distribution: the paired test; non-Gaussian distribution: Wilcoxon test). Significance difference Before vs After * *p* < 0.05; Abbreviations: SOD—superoxide dismutase, TAS—total antioxidant status, ROS—reactive oxygen species.

**Figure 3 antioxidants-10-01018-f003:**
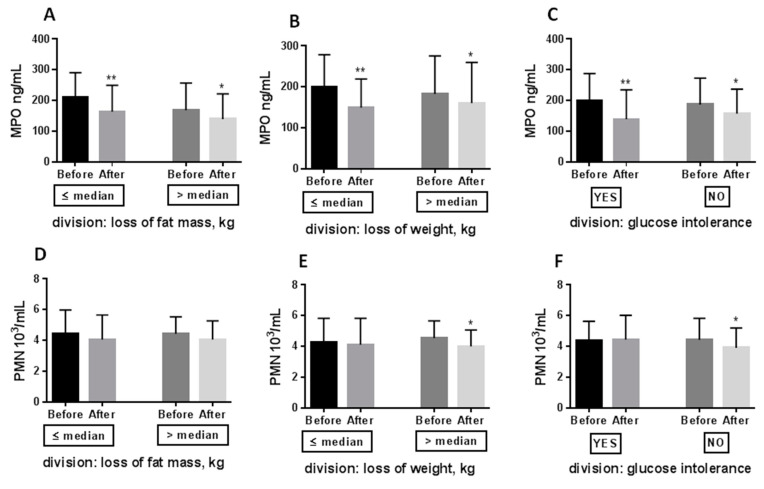
Effect of 8 weeks caloric restriction on serum myeloperoxidase activity (**A**–**C**) and neutrophile count (**D**–**F**). Patients after caloric restriction were divided according to the median value regarding the fat mass loss, the bodyweight loss, and the glucose concentration after 2 h of OGTT at baseline. The data were analyzed using the paired/unpaired *t*-tests depending on Gaussian/non-Gaussian distribution. Significance difference Before vs. After: * *p* < 0.05; ** *p* < 0.01. Abbreviation: MPO—myeloperoxidase, PMN—polymorphonuclear leukocytes.

**Figure 4 antioxidants-10-01018-f004:**
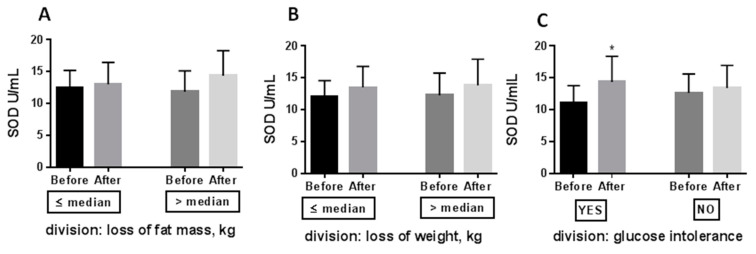
Effect of 8 weeks caloric restriction on serum superoxide dismutase (panel **A**–**C**). Patients after caloric restriction were divided according to the median value regarding the fat mass loss, the bodyweight loss, and the glucose concentration after 2h of OGTT at baseline. The data were analysed using the paired/unpaired *t*-tests depending on Gaussian/ non-Gaussian distribution. Significance difference Before vs. After: * *p* < 0.05. Abbreviations: SOD—superoxide dismutase.

**Table 1 antioxidants-10-01018-t001:** The inclusion and exclusion criteria for the study.

Inclusion Criteria	Exclusion Criteria (Occurrence of at Least One of the Following)
Age: >18 years BMI > 30 kg/m^2^ Increased adipose tissue content, measured by BIA Expresing informed consent to particitate in the study	Diabetes mellitus Acute coronary syndrome over past 6 month Previous or current neoplastic disease (receiving radiotherapy, chemotherapy) Cardiovascular, autoimmune, congenital metabolic or liver diseases Inflammatory bowel disease (Crohn’s disease, ulcerative colitis) Bariatric surgery Change in body weight greater than 2 kg over the past 3 months Pregnancy Vegetarian (or other alternative) diet Eating disorders (anorexia, bulimia), Mental disorders Alcohol/drug abuse Ongoing antibiotic therapy, steroid therapy

Abbreviations: BMI—body mass index, BIA—electrical bioimpedance.

**Table 2 antioxidants-10-01018-t002:** Baseline characteristics of study subjects.

Parameters	
Age, years	37.7 ± 11.3 (min: 20, max: 65)
Number of patients, *n*	53
Female, *n* (%)	38 (72)
Morbid obesity BMI > 40, *n* (%)	17 (32)
Smoking, *n* (%)	16 (30)
Glucose intolerance, *n*(%)	15 (28)
Metformin treated, *n* (%)	24 (45)
Treated for hypertension, *n* (%)	15 (28)
Treated for thyroid disease, *n* (%)	10 (19)

**Table 3 antioxidants-10-01018-t003:** Diet composition.

Diet Composition	% of Total Energy Intake
Carbohydrates	50–55
Complex carbohydrates Saccharose	45–50 <10
Protein	20–25
Fat	25
Saturated fatty acids Monounsaturated fatty acids Polyunsaturated fatty acids	7 10 8
Cholesterol intake (mg/day)	<300

**Table 4 antioxidants-10-01018-t004:** Characteristics of patients before and after eight weeks of calorie restriction diet. The comparisons were made using the *t*-tests (Gaussian distribution: the paired test or the unpaired test; non-Gaussian distribution).

Parameters	before	after
Body mass, kg	109.4 ± 20.3	103.5 ± 19.7 ****
BMI, kg/m^2^	37.8 ± 5.9	35.7 ± 5.8 ****
Morbid obesity BMI > 40 *n*(%)	17 (32)	8 (15) *
WHR	1.7 ± 0.2	1.7 ± 0.2
WC, cm	114.4 ± 14.3	111.3 ± 15.2 ****
Fat mass, kg (%)	46.1 ± 13.9	41.4 ± 13.4 ****
SBP, mmHg	128 ± 11	122 ± 11 ***
DBP, mmHg	81 ± 9	80 ± 8
HR, beats/min	73.7 ± 11.2	72.4 ± 10.6
Cholesterol, mg/dL	196.4 ± 36.4	189.9 ± 41.7 *
TG, mg/mL	154.1 ± 93.6	129.7 ± 72.9 *
LDL	121.5 ± 49.8	114.1 ± 36.1
HDL	50.8 ± 14.3	50.9 ± 12.7
Hb, g/dL	14.3 ± 1.1	14.3 ± 1.2
Glucose, mg/dL	95.8 ± 9.6	95.7 ± 14.9
Insulin, mU/mL	16.8 ± 9.9	16.2 ± 14.4
HOMA-IR	4.0 ± 2.5	3.4 ± 2.0 *
Leptin, ng/mL	53.5 ± 29.3	40.5 ± 26.3 ****
Adiponectin, µg/mL	1.9 ± 1.1	2.0 ± 0.9
Vaspin (Serpin A12), ng/mL	312.6 ± 368.5	247.8 ± 293.7 ***
Rezistin, ng/mL	16.2 ± 9.1	15.5 ± 9.0
hs CRP, µg/mL	7.4 ± 5.8	5.5 ± 4.4 ****
hs Il-6, pg/mL-	2.0 ± 1.4	1.23 ± 1.0 ****
hs TNF- alpha, pg/m	17.1 ± 8.2	8.4 ± 8.8 ****

* indicates significant differences before and after intervention with * *p* ≤ 0.05, *** *p* ≤ 0.001, **** *p* ≤ 0.0001. Abbreviations: BMI—body mass index, WHR—waist/hip ratio, WC—waist circumference, SBP—systolic blood pressure, DBP—diastolic blood, HR—heart rate, TG—triglycerides, LDL—low-density lipoproteins, HDL—high-density lipoproteins, HOMA-IR—homeostatic model assessment of insulin resistance (fasting insulinemia (mU/mL) × fasting glycemia mg/dL/405), CRP—C-reactive protein, Il-6—interleukin 6, TNF- alpha —tumor necrosis factor- alpha, hs—high sensitive assay.

## Data Availability

The data supporting the conclusions of this article are included within the manuscript. The dataset is available from the corresponding author on request.

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
