# Peer review of "Moderate Caloric Restriction Partially Improved Oxidative Stress Markers in Obese Humans"

_antioxidants, 2021, doi:10.3390/antiox10071018_

Round 1

Reviewer 1 Report

Kanikowska et al study the effect of short-term moderate caloric restriction in anthropometric and biochemical parameters, including oxidative stress markers, of a group of obese subjects. They found MPO and SOD affected by the CR, as well as inflammatory markers and adipokines, leading to a moderate improvement of the obese phenotype. The study is well conducted; however, some questions need to be addressed:  

Major:    

-What is the scientific reason to exclude 5 subjects that follow the CR program but increase weight?   

- “A similar relationship was not documented for SOD (data not shown)”. What do the authors mean by this? Please include the data of SOD.   

-How were the in vitro experiments performed? The serum was a pool from different subjects? How many times the experiment was replicated? Also, show the results of the controls in the graph.   

-MPO is secreted mainly by neutrophils and monocytes. Nevertheless, MPO levels and neutrophil count are affected by different criteria: MPO is affected in all the cases including fat loss but neutrophil count only under higher weight loss and normoglycemic subjects but not dependent on fat loss. Although the authors attempted to discuss the reason behind this, the mechanism leading to the decrease in MPO is not clear. Did the authors examine the effect of metformin supplementation in the studied parameters? -Are there differences between males and females?  

-The authors comment on the significance of this intervention for public health, but with the data provided only an immediate effect can be proved. As state in the introduction long-term interventions tend to fail and subjects regain the weight. Apart from the effect observed in the study, after the 8 weeks of CR did the metabolic parameters remain lower? What are the long-term benefits of short-term CR?  

Minor:    

-Line 42- please correct “defence”.   

-Line 83-84. These abbreviations seem out of place: STEMI- ST-elevation myocardial infarct, NSTEMI- Non-ST-elevation myocardial infarct.   

-Line 225: please include a definition for PMN. 

Reviewer 2 Report

The authors analyzed if short-term calorie restriction would have impact on oxidative stress markers in humans. Therefore, they performed a single group intervention and analyzed metabolic effects as well as various oxidative stress and inflammation markers in serum samples before and after the calorie restriction phase. Additionally, generation of reactive oxygen species was measured in endothelial cells in vitro.

The manuscript deals with an important topic and is well readable, however, some points should be addressed:

Abstract:

line 20: please explain ROS and all following abbreviations (line 26)

line 24: decrease in body weight (by 5.9±4.6) --> please add kg

Materials and Methods

Type 2 diabetes was an exclusion criterion, however, introduction of metformin was allowed. Please clarify.

Maybe the decision to dispense a control group limits the meaningfulness.

Results

Table 4: Please clarify if the HOMA-IR is correctly calculated in the post CR condition

Table 4 shows a trend for reduced levels of fasting insulin, although the difference was not statistically significant.  Since insulin is an important regulator of weight loss, authors might additionally prove and discuss if reduction of fasting insulin levels (according to loss of fat mass, loss of weight or glucose tolerance in Fig.3) might be associated with improvements in oxidative stress.

Discussion

line: 266: Did the authors really mean: 'A primary focus of our work was to determine if calorie restriction diets would decrease oxidative stress capacity.' Shouldn't it rather be: 'A primary focus of our work was to determine if calorie restriction diets would decrease oxidative stress.' or 'A primary focus of our work was to determine if calorie restriction diets would increase oxidative stress capacity.

line 329: The authors wrote: 'Reducing body weight and fat mass decreases pro-inflammatory adipokines, lipid parameters, and insulin sensitivity'. Shouldn't it rather be: '... and insulin resistance' or '... and increases insulin sensitivity'?

Please overwork the manuscript focussing on the exactness of the meaning.

Maybe a native speaker should read the manuscript, there are several typos etc. For example:

line 267: The antioxidant effect of the dietary intervention was measure. --> measured

Round 2

Reviewer 1 Report

All questions have been addressed